# Correlation of pN Stage and Hypoechogenicity with Tumour Encapsulation and Vascular Invasion in Thyroid Cancer (TC): A Comprehensive Analysis and Clinical Outcomes

**DOI:** 10.3390/cancers16112019

**Published:** 2024-05-26

**Authors:** Krzysztof Jurkiewicz, Michał Miciak, Szymon Biernat, Beata Wojtczak, Krzysztof Kaliszewski

**Affiliations:** Department of General, Minimally Invasive and Endocrine Surgery, Wroclaw Medical University, 50-367 Wrocław, Poland; michal.miciak@student.umw.edu.pl (M.M.); szymon.biernat@student.umw.edu.pl (S.B.); beata.wojtczak@umw.edu.pl (B.W.); krzysztof.kaliszewski@umw.edu.pl (K.K.)

**Keywords:** thyroid cancer management, TNM staging, tumour encapsulation, tumour vascular invasion, hypoechogenicity, thyroid surgery

## Abstract

**Simple Summary:**

This study aimed to explore the relationship of the pN stage of thyroid cancer (TC) and ultrasound hypoechogenicity with encapsulation and vascular invasion. A retrospective study involving 678 TC patients revealed that a greater pN stage was correlated with encapsulation and vascular invasion as well as with positive hypoechogenicity. These findings have significant prognostic and clinical implications for managing TC. Discovering these correlations confirms the importance of the TNM scale in stratifying TC patients based on disease severity. The results suggest that the hypoechoic patterns observed on thyroid ultrasound may serve as indicators of aggressive tumour behaviour and increased invasion risk. This analysis highlights diverse invasion patterns among TC patients, which may have significant clinical implications for diagnostic processes and treatment strategy selection. Further research is necessary to validate these findings and explore new biomarkers and imaging methods to refine TC staging systems and improve TC patient care.

**Abstract:**

In this retrospective study, the relationship between the pN stage of TC and the ultrasound hypoechogenicity of tumour encapsulation and vascular invasion was investigated. The data of a total of 678 TC patients were analysed. The goal of this study was to assess the significance of the pTNM score and preoperative ultrasound features in predicting cancer prognosis and guiding therapeutic decisions in patients with TC. The main research methods included a retrospective analysis of patient data, mainly the pTNM score and presence of tumour encapsulation and vascular invasion obtained from histopathological results and preoperative ultrasound imaging. Patients with well-differentiated TCs (papillary and follicular) were extracted from TC patients to better unify the results because of similar clinical strategies for these TCs. Significant associations were observed between advanced pN stage and the presence of encapsulation and vessel invasion. The majority of pN1a patients exhibited encapsulation (77.71%; *p* < 0.0001) and vascular invasion (75.30%; *p* < 0.0001), as did the majority of pN1b patients (100%; *p* < 0.0001 and 100%; *p* < 0.0001, respectively). Less than half of the patients with hypoeghogenic patterns presented with encapsulation (43.30%; *p* < 0.0001) and vascular invasion (43.52%; *p* < 0.0001), while the vast majority of patients without hypoechogenicity did not present with encapsulation (90.97%; *p* < 0.0001) or vascular invasion (90.97%; *p* < 0.0001). Hypoechogenicity was found to be indicative of aggressive tumour behaviour. The results of this study underscore the importance of accurate N staging in TC and suggests the potential use of ultrasound features in predicting tumour behaviour. Further research is needed to confirm these findings and explore additional prognostic markers to streamline TC management strategies and improve patient outcomes.

## 1. Introduction

The complex interaction between tumour characteristics and tumour staging is critical for the successful treatment of thyroid cancer (TC). Recent advances in oncology research have emphasized the importance of refining staging parameters to better understand the nuanced features of TC and target-modified treatment strategies. Particularly important is the pN stage in the pTNM scoring system [1,2]. The evolution of TC staging has been propelled by a number of factors, including advances in diagnostic imaging, clarification of the molecular pathways involved in carcinogenesis, and improvements in surgical techniques. Under this background, the following question remains to be asked: how is the pN stage related to tumour encapsulation and vascular invasion, and how does this affect clinical decision-making [3]? Based on an increasing body of relevant literature and applying insights from clinical practice, the goal of this study was to investigate the relationship between pN stage and other histopathological features of TC. By summarizing the results from various sources, including PubMed and other reputable medical databases, we aimed to clarify the impact of tumour encapsulation and vascular invasion on disease progression and treatment outcomes. Crucial to this study is the recognition of TC as a heterogeneous medical condition involving different histological subtypes with varying clinical behaviours and prognostic implications. From relatively less aggressive papillary thyroid carcinoma (PTC) and follicular thyroid carcinoma (FTC) to more aggressive medullary (MTC) and anaplastic variants, TC presents considerable diversity in terms of its biological behaviour and response to therapy [4]. PTC and FTC are perceived as well-differentiated thyroid carcinomas. They are also the most common, comprising 80–84% and 6–10% of all TCs, respectively. More than 90% of all well-differentiated TCs can now be treated by surgery combined with radioactive iodine therapy [5]. The treatment strategy is already changing to a more aggressive one (chemotherapy, total profilactic thyroidectomy with genetic burden), for instance, in the case of MTC [6]. Furthermore, the development of precision medicine has revolutionized the management of TC, initiating an era of personalized treatments adapted to individual tumour characteristics. In this context, understanding the correlation between the pTNM score and tumour infiltration pattern would have great clinical significance, offering insights into disease aggressiveness and guiding therapeutic decisions [7]. Another aspect worth discussing is the fact of preoperative diagnosis of TC. Fine-needle aspiration biopsy (FNAB) or thyroid ultrasonography holds a major role at this point. Hypoechogenicity in thyroid ultrasonography refers to the tissue’s ability to reflect ultrasound waves to a lesser degree than surrounding structures. In practice, this means that hypoechoic tissue appears darker on ultrasound images than surrounding tissues. It is an indicator used to evaluate a variety of anatomical structures and pathological changes in the body. The evaluation of hypoechogenicity is always performed in the context of comparison with other tissues, most commonly muscle, which have moderate echogenicity [8,9]. Hashimoto disease, or chronic lymphocytic thyroiditis, is often associated with characteristic changes in thyroid echogenicity. In this disease, the thyroid parenchyma outside of the nodules is often hypoechoic, the result of inflammatory infiltration and fibre negativity. This makes hypoechogenicity of the entire thyroid gland in Hashimoto’s a non-specific indicator, as inflammation can cause decreased echogenicity in the entire thyroid organ [10]. However, hypoechogenicity of thyroid nodules is a more significant diagnostic parameter. Hypoechoic nodules may indicate malignancy, especially when accompanied by other features such as microcalcifications, nonregular borders, or increased vascular flow within the nodule. Studies indicate that, despite the general hypoechogenicity of the thyroid gland in Hashimoto’s, nodular lesions with lower echogenicity still have a high predictive value in the diagnosis of TC [11,12]. Recent studies confirm that the hypoechogenicity of thyroid nodules remains an important parameter in the differential diagnosis, even in the context of Hashimoto’s disease. A study by Lee et al. demonstrated that the hypoechogenicity of a nodule, in combination with other ultrasound risk cues, significantly increases the probability of detecting TC [13]. Moreover, according to a meta-analysis by Wang et al., nodule hypoechogenicity in Hashimoto’s is still an important indicator that does not lose its predictive value despite the overall hypoechogenicity of the thyroid parenchyma in this disease [14]. Explaining the relationship between N stage and tumour capsule/vascular invasion, we aimed to better understand the biology of the disease and to inform more precise prognosis and treatment strategies for the management of TC. We further performed an analysis between N stage and thyroid hypoechogenicity on preoperative ultrasound to see how such a pattern already provides a chance at early diagnosis to predict whether lymph nodes may be affected. The presence of vascular invasion and infiltrating capsular lesions in the histopathologic evaluation of malignant tumours has a significant impact on patient prognosis. These features are often associated with the aggressive nature of the tumour and an increased risk of recurrence [15,16,17,18]. However, our study focused on analysing the correlations between these factors, which may be crucial for the treatment of TC patients. Identifying these correlations may provide valuable clinical insights, allowing physicians to monitor patients more closely and customize therapy to a more aggressive approach. Through a comprehensive review of the literature and empirical analysis, this study aims to contribute to the ongoing discourse on TC staging and management practices.

## 2. Materials and Methods

This retrospective analysis involved analysing the records of 5806 patients who were consecutively surgically treated for single and multiple thyroid nodules (TNs) between January 2008 and December 2022 in the Department of General, Minimally Invasive, and Endocrine Surgery at Wroclaw Medical University. After the application of patient inclusion and exclusion criteria, 678 (11.7%) patients were included in the study cohort. The selection process is presented in a flow diagram in Figure 1.

The study was approved by the Institutional Review Board and Ethics Committee of Wroclaw Medical University, Wroclaw, Poland. All of our patients provided informed consent at admission, which indicated that the results may be used for research purposes. The data were analysed retrospectively and anonymously from established medical records. The authors did not have direct access to the study participants.

From the initial search, 678 patients with malignant thyroid tumours were selected. Among the patients, there were 581 women and 97 men, with a mean age of 52.9 ± 16.5 years. All patients lived in geographically iodine-sufficient areas prior to surgery and underwent fine-needle aspiration biopsy (FNAB) before surgical intervention. Patients with malignant thyroid tumours always underwent therapeutic and diagnostic lymphadenectomy of the mid-cervical compartment. Both cytological and postoperative histopathological samples were reviewed by an experienced pathologist, and final diagnoses were classified according to the Bethesda System for Reporting Thyroid Cytopathology and the World Health Organization’s classification of thyroid malignancy [19]. All patients were evaluated, and the following parameters were recorded: sex, age, tumour size, tumour shape, echogenicity, pTNM stage, vascularization, microcalcifications, and tumour type (single, multifocal, bilateral).

### Statistical Analysis

Regarding statistical analysis, numerical and percentage values were calculated for qualitative variables, and means and standard deviations were calculated for quantitative variables. Group comparisons were made using the chi-square test for qualitative variables and Student’s t test where appropriate. A significance threshold of *p* < 0.05 was applied for all the statistical analyses, which were performed using Statistica 10.0 software (StatSoft Inc., Tulsa, OK, USA).

## 3. Results

In total, 678 of 5806 patients (11.7%) were diagnosed with malignant thyroid tumours that met all the diagnostic criteria. The participants encompassed a heterogeneous spectrum of thyroid malignancies, including PTC, FTC, MTC, undifferentiated thyroid carcinoma (UTC), sarcoma, lymphoma, squamous cell carcinoma, myeloma, and secondary tumours, covering a 14-year period from 2008 to 2022. Table 1 provides an overview of the demographic and clinical characteristics of the entire TC cohort, as well as two distinct subgroups: PTC and other types of TC.

Among the TC patients, there were 581 (85.69%) women and 97 (14.31%) men, and the mean age at diagnosis was 51.66 ± 15.98 years (range 18–81 years). Among the various TC subtypes, PTC was the most common (579 patients, 85.39%), followed by FTC (31 patients, 4.57%) and MTC (24 patients, 3.53%). There were significant discrepancies in terms of age, surgical procedures, reoperation rates, and pTNM stage between patients with PTC and those with other TC subtypes. Notably, the mean age at diagnosis was significantly lower in patients with PTC than in patients with other TC types (*p* < 0.0001). Total thyroid resection was more frequently performed for PTC patients than for patients with other TC subtypes (*p* < 0.0001), with radical surgery being the main procedure for PTC patients. Conversely, patients with TC other than PTC often required reoperation due to nonradical primary procedures (*p* = 0.002). In addition, patients with PTC were more likely to have pTNM stage I, while patients with other TC subtypes were more likely to have pTNM stage IV (*p* < 0.0001). In the group of TC patients without PTC, significantly greater percentages of patients with advanced pathological categories (pT4b, pN1b, and pM1) were observed than in the subgroup of patients with PTC (*p* < 0.0001 for all). Ultrasound-guided (UG)-FNAB results were available for all patients (100%). The selected ultrasound features differed significantly among the total TC group, the PTC group, and the other TC group (*p* < 0.05 for all). In particular, patients with TC other than PTC were more likely to present with a tumour greater than 5 mm long, an irregular tumour shape, hypoechogenicity, microcalcifications, and increased vascularization than PTC patients. At the very end, from the group of all TC patients, we separated a group of patients with well-differentiated TC (PTC and FTC) for further analysis. These types of TC have a similar course, treatment, and prognosis in contrast to the more difficult to manage MTC or sarcoma for which the approach is more aggressive from the outset.

### 3.1. Tumour Encapsulation and Vascular Invasion

Analysis of thyroid encapsulation and vascular invasion among well-differentiated TC patients revealed different results. Of the total cohort, 385 (62.95%) patients were classified as pN0. Within this group, 142 (36.98%) patients showed invasion of the thyroid capsule, while 242 (63.02%) showed no evidence of invasion. A total of 210 (54.69%) pN0 patients presented with vascular invasion, while 174 (45.31%) showed no evidence of vascular invasion. Among the 166 pN1a patients (27.21%), 129 (77.71%) also exhibited invasion of the thyroid capsule and 125 (75.30%) exhibited vascular invasion. Among pN1b patients, 31 (5.09%) had both thyroid capsule invasion and vascular invasion. Additionally, 29 (4.75%) patients were classified as pNx, among whom 13 (44.83%) showed thyroid capsule invasion, and 13 (44.83%) showed vascular invasion. The results are presented in Table 2 and Figure 2.

### 3.2. Hypoechogenicity

The presence of hypoechogenicity on thyroid ultrasound among well-differentiated TC patients was also analysed with respect to tumour encapsulation and vascular invasion. The results are presented in Table 3.

Data analysis revealed that hypoechogenicity was common among well-differentiated TC patients (observed in 74.59%). Encapsulation was present in 43.30% of these patients, but only in 9.03% of patients without hypoechogenicity. These results suggest that hypoechogenicity may be an indicator of a greater tendency for encapsulation. Similarly, vascular invasion was more frequent in patients with hypoechogenicity (43.52%) than in those without hypoechogenicity (9.03%). This indicates a potential correlation between hypoechogenicity and vascular invasion in TC patients. The results are presented in Figure 3.

## 4. Discussion

The results of our study provide valuable insights into the complex relationship between pTNM score, hypoechogenicity, and tumour encapsulation and vascular invasion in TC. Our findings emphasize the clinical importance of the pN stage as a predictor of tumour behaviour and aggressiveness, resulting in better disease management and prognosis. The majority of well-differentiated TCs have an indolent course and a very good prognosis [20]. Therefore, in our study we did not focus on 5-year survival as a prognostic oncological indicator. By proving the correlation between pN and tumour encapsulation and vascular invasion, we are able to plan further clinical decisions (aggressive approach, frequency of follow-up visits after thyroidectomy, qualification for radioiodine follow-up treatment, or qualification for complementary lymphadenectomy) while having any of the studied features in the absence in the histopathologic result, for example pNx. The value of our study is purely clinical. If a patient presents one of the analysed features then we could predict the other based on a positive correlation. Data analysis uncovered varying patterns of invasion among TC patients, which may have significant clinical implications for diagnostic processes and the selection of treatment strategies. For well-differentiated TCs, surgical treatment is usually sufficient, with additional radioiodine treatment in cases of high risk of recurrence [21,22]. Thus, it may be lymph node metastasis (pN1), and from our analysis we would think of complementary treatment in such patients. In the case of TCs with a worse prognosis and higher mortality (such as MTC), the post-treatment is more aggressive from the start and consists of a total therapeutic thyroidectomy and adjunctive pharmacotherapy (tyrosine kinase inhibitors). The thyroidectomy procedure is almost always accompanied in this case by central only or central and lateral neck dissection [23]. Therefore, we decided to exclude TCs other than well-differentiated TCs from our analysis at the end, because in their cases the strategy from the beginning is very radical. The classification of TC has undergone many changes over the years. For example, the WHO currently divides FTC into additional subcategories that differ in their potential for invasiveness. This division is based precisely on the presence of capsular infiltration and angioinvasion [24]. Among the independent prognostic factors for a worse prognosis in TC patients, the authors of a previous study emphasized distant metastases, older age, and lymph node involvement (N1) [25]. Other studies have concluded that the procedure of choice for patients with even mild thyroid disease but increased suspicion of TC should be total thyroidectomy [26]. An accurate histopathological evaluation of many of the features we evaluated in our study is possible only after the surgical procedure. The observed association between pN stage and other histopathological features, such as tumour capsularity and vascular invasion, provides compelling evidence for the utility of pTNM staging in stratifying TC patients based on disease stage. In particular, our data revealed a progressive increase in the incidence of thyroid capsule involvement and vascular invasion in higher pN stages, particularly among patients classified as pN1b. These findings underscore the prognostic value of pN stage in predicting the extent of tumour spread and its impact on disease progression.

In our study, we noted a strong correlation between pN stage and the incidence of tumour encapsulation. Akbulut et al. also observed similar results, finding a greater percentage of lymph node metastases in patients with tumour capsular invasion (36.8% metastasis for invasion and 30.4% for no invasion) [27]. Carcangiu et al. noted a higher rate of not only lymph node metastasis but also distant metastasis to the lungs (M1) in patients with capsular invasion [28]. Rivera et al. also noted the significance of capsular invasion of noninvasive PTC; despite the low metastatic potential of the tumour, up to 27% of patients with capsular invasion developed lymph node metastases [29]. In contrast, FTC is widely believed to exhibit very indolent behaviour, even when only the capsule is involved [30]. Jung et al. also reported that the presence of a positive tumour margin, which is naturally the capsule, is a strong predictor of the presence of lymph node metastases [31]. It is also worth noting that by determining the cN1 stage, it is possible to plan appropriate surgical treatments, even before surgery. Central compartment lymphadenectomy is a well-accepted treatment for cN1a stage TC, and additional lateral compartment lymphadenectomy is a well-accepted treatment for cN1b stage TC. The role of lymphadenectomy in the prevention of dissemination in cN0 TC patients has not yet been clarified [32]. Lymphadenectomy, together with total thyroidectomy and supplemental radioiodine administration, is an appropriate radical treatment option for patients with tumours with certain characteristics, such as extrathyroid invasion, lymph node involvement, multifocal tumours, or capsular invasion [33]. Considering these features, it should be noted that the use of radioiodine in particular should be limited to invasive TCs. Studies have demonstrated the emerging manifestations of overtreatment from this unique form of targeted therapy [34]. In conclusion, as seen in our study, the relationship between capsular infiltration and the pN stage is associated with a high tumour metastatic potential. Thus, the surgical approach should be more radical (e.g., in patients with cervical lymph node metastases), that is, lymphadenectomy, or more frequent follow-up should be conducted after surgery [35]. Capsular infiltration is also sometimes difficult for pathomorphologists themselves to interpret. There is a general consensus that capsular invasion is diagnostically equivalent to malignancy. However, determining the degree of capsular invasion involves subjective evaluation of the specimen. Thus, for correct application of this feature in clinical practice, collaboration with specialists evaluating many thyroid preparations per year is needed [36]. There are also reports of the use of less classical TC treatments, such as microwave ablation, for tumours demonstrating capsular invasion. In a study by Wu et al., patients with TC N0 with positive capsular infiltration were successfully treated; no local recurrence was observed in any patient, and the method was evaluated as effective and safe [37]. Thus, it is important to take all features into consideration during clinical planning and not to focus mainly on the examined capsular infiltration.

The incidence of vascular invasion also increased dramatically with pN stage in our study. A fully analogous relationship was described in a publication by Reilly et al. The authors concluded that in PTC, the only TC type they studied, there is a clear positive relationship between tumour vascular infiltration and disease aggressiveness, necessitating a more radical approach to therapy and closer follow-up of patients [38]. According to the College of American Pathologists (CAP) and International Collaboration on Cancer Reporting (ICCR) guidelines, the extent of vascular invasion is stratified as focal (1–3 foci of invasion) or extensive (more than 4 foci). TCs of the extensive type are considered to have high risk of recurrence. For such tumours, an aggressive therapeutic approach is recommended, including the use of radioactive iodine ablation [39]. Fonseca et al. also noted that the presence of angioinvasion is associated with a greater rate of lymph node involvement (N1 stage) and distant metastases (M1 stage). Angioinvasion is therefore considered an unfavourable prognostic factor in TC patients, justifying a more aggressive therapeutic approach and more frequent follow-up in such patients, which is also consistent with the results of our study [40]. However, it should be noted that extensive angioinvasion is also an independent predictor of local tumour recurrence in addition to lymph node spread. Xu et al. studied a cohort of 276 patients with FTC and demonstrated a similar relationship [41]. Ghossein et al. also noted that angioinvasion present in four or more tumour foci was the most important predictor of recurrence in patients with the relatively rare Hürthle cell carcinoma (HCC) with respect to patients with angioinvasion in fewer than four foci [42]. In their analysis, Ito et al. noted that angioinvasion independently predicted distant recurrence relative to other parameters (older age, male sex). In contrast, even extensive capsular invasion by the tumour had no significant effect on recurrence, and thus the authors suggested that the definition of wide capsular infiltration should be changed [43]. The literature differs not only in terms of outcomes but also in the inclusion criteria and classification of thyroid tumours. Yamazaki et al. noted that studies have included different histologic types of thyroid cancer (PTC, FTC, and others) in evaluating angioinvasion. Comparisons of previous studies are also complicated by the nature and incidence of HCC-type tumours. The cohorts are limited to a small number of patients, which does not allow adequate conclusions to be drawn [44,45]. It remains worth mentioning that lymph-node involvement and activation of immune system may promote vascular invasion in TC through the production of vascular endothelial growth factor (VEGF). Gulcelik et al. reported higher VEGF expression in adjacent non-tumoral tissue in PTC patients with lymphocytic thyroiditis than in those without the condition. This finding suggests that VEGF may play a role in the tumour microenvironment, affecting TC progression. In clinical practice, monitoring VEGF expression in adjacent non-cancerous tissue could help identify patients at higher risk of aggressive TC, which could lead to more targeted and intensive treatment. The importance of VEGF as a major mediator of angiogenesis in TC is strongly emphasized, with high VEGF expression correlating with greater aggressiveness and worse prognosis. This finding confirms the importance of VEGF as a potential therapeutic target [46]. In addition, Tian et al. found that VEGF-C protein expression was significantly more frequent in PTC with lymph node metastasis than in cases without metastasis. In clinical practice, this implies that VEGF-C could act as a prognostic biomarker for the risk of lymph node metastasis, which could influence decisions on surgery and the use of additional therapies. The results show that inhibitors such as sorafenib and lenvatinib significantly improve disease control and patient survival. In clinical practice, these therapies can inhibit angiogenesis, reducing tumour growth and metastasis, offering patients with advanced TC better prognosis and quality of life [47]. Monitoring VEGF and VEGF-C levels can serve as important prognostic biomarkers, while VEGF-TKI pathway inhibitors offer a novel and effective therapeutic approach for treating advanced TC. Thus, further studies are needed to adequately assess the relationship between TC angioinvasion and lymph node metastasis according to the different histological types of TC and possible biomarkers.

The correlation between hypoechogenicity detected on thyroid ultrasound and tumour capsular and vascular infiltration further strengthens our understanding of the biological behaviour of TC. Our analysis revealed a significant association between the presence of hypoechogenicity and invasion of both the thyroid capsule and blood vessels, suggesting that hypoechogenicity may serve as a proximal marker of aggressive tumour behaviour and increased risk of invasion. Given the lack of hypoechogenicity and other clinical and ultrasound data (age less than 55 years, tumour size less than 5 mm, no microcalcifications, no irregular tumour shape, and no blunt edges), we would be inclined to predict a low risk of metastasis and local recurrence, allowing for a less aggressive clinical approach [48]. On the other hand, identifying the relationship between N stage and hypoechogenicity on ultrasound imaging may prompt clinicians to adopt a more vigilant approach to disease monitoring and future treatment planning. Such patients will benefit from closer monitoring and more aggressive therapeutic interventions to reduce the risk of disease recurrence and metastasis. An analysis by Lim et al. showed that significant lesion hypoechogenicity was correlated with aggressive tumour histological features (tumour growth pattern and tumour fibrosis). This can be translated into clinical features, as tumours displaying the aforementioned features were associated with more frequent lymph node metastasis than those not displaying aggressive histological features and hypoechogenicity [49]. Jiwang et al. proved that the consideration of certain ultrasound features (including the absence of hypoechogenicity) can improve the prediction of TC metastasis to lymph nodes [50]. The results of Prieditis et al. also demonstrated the correlation of a low risk of malignancy of TCs with a hypoechoic pattern [51]. Thus, the correlation between hypoechogenicity and TC aggressiveness revealed among our results conforms with the findings in the literature. However, it cannot be concluded that any ultrasound criterion can effectively exclude or confirm malignancy [52]. Hypoechogenicity also appears to be a helpful feature in planning biopsy therapeutic management. In cases of cytologic ambiguity and concomitant nodule hypoechogenicity, a more conservative approach and repeat biopsy for a better result are warranted [53]. As in the case of angioinvasion, ultrasound features also appear to vary depending on the TC subtype. The ultrasound features of FTC that raise a suspicion of malignancy are different from those of PTC. Instead of hypoechogenicity, the most important feature associated with an increased risk of FTC malignancy is capsular protrusion, followed by the presence of calcification [54]. In conclusion, each of the parameters studied in our analysis should be examined together with the entire clinical and pathomorphologic picture, and clinical decisions should be made with respect to a complete overview of the disease. Analysis of these data underscores the diverse patterns of invasion among patients with TC, which may have important clinical implications in diagnostic processes and the choice of treatment strategies.

Our study highlights the evolving landscape of TC management in the era of precision medicine. By clarifying the relationship between pTNM stage and tumour behaviour, we provide clinicians with valuable insights into the biological basis of TC and clarify how personalized treatment strategies can be developed and adapted to individual patient profiles. This paradigm shift towards precision oncology holds promise for improving patient outcomes and optimizing resource allocation in the treatment of TC. Future perspectives of this study would be to conduct a prospective study including a larger cohort of patients with hypoechoic thyroid lesions in the overall analysis, or to create a group of patients with a positive correlation of the studied features and analyse the effectiveness of therapy after incorporating the aggressive approach described in the study.

Our paper has some limitations. First, it is a retrospective analysis, so some inaccuracies typical of such studies were unfortunately present. Second, this study was observational and performed in only a single institution. Third, one of the characteristics for this study was obtaining histopathological results, so the study contained a selection bias be-cause there were only evaluated patients with malignant tumours who underwent thy-roidectomy. Finally, a relatively small number of patients were analysed in this study.

## 5. Conclusions

In summary, our study underscores the importance of pN stage in predicting tumour encapsulation and vascular invasion in TC. By providing clinicians with crucial prognostic information, we can help them make informed therapeutic decisions and optimize patient outcomes. Further research is necessary to validate our findings and explore novel biomarkers and imaging modalities to refine TC staging and improve patient care.

## Figures and Tables

**Figure 1 cancers-16-02019-f001:**
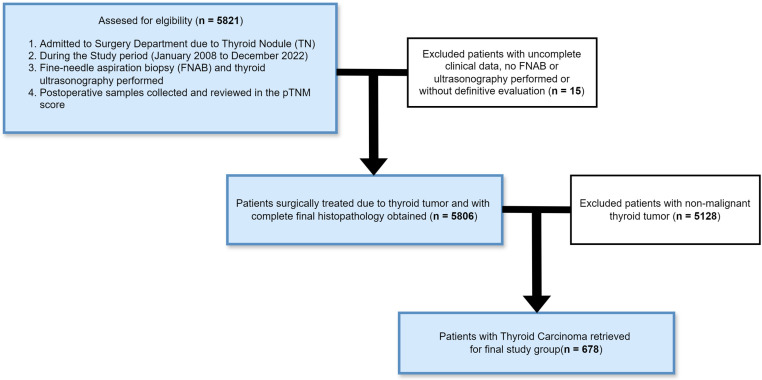
Study group selection from the records of 5821 patients admitted to the Surgery Department during the study period. All selected patients underwent fine-needle aspiration biopsy (FNAB), thyroid ultrasonography, and thyroid surgery and received histopathology results. The results were evaluated in terms of TNM score, and tumour encapsulation, vascular invasion, and microcalcifications were examined.

**Figure 2 cancers-16-02019-f002:**
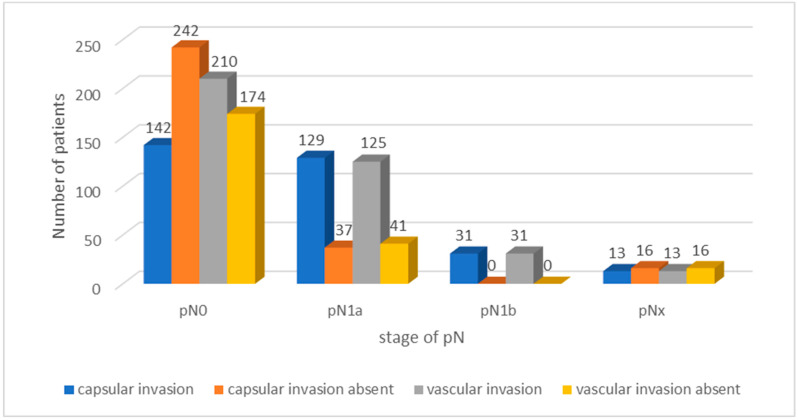
Analysis of tumour capsular and vascular invasion as a function of pN stage in well-differentiated TC.

**Figure 3 cancers-16-02019-f003:**
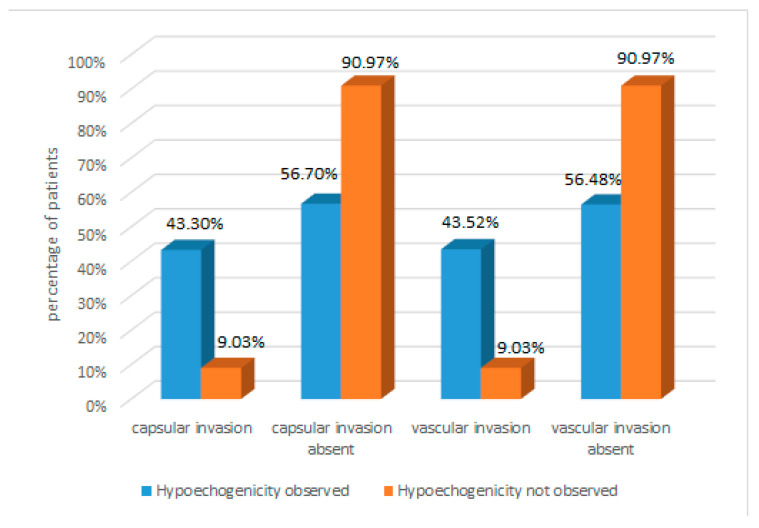
Analysis of capsular and vascular invasion according to hypoechogenicity in well-differentiated TC.

**Table 1 cancers-16-02019-t001:** Demographic and clinical characteristics of patients with TC. The *p*-values were calculated using Pearson’s correlation test, where *p* < 0.0001 indicates a statistically significant correlation between the variables.

Feature		Number of Patients	*p* Value
Sex	Male	97 (14.31%)	0.133
	Female	581 (85.69%)	
Age (years)	<55	385 (56.78%)	<0.0001
	>55	293 (43.22%)	
Type of surgery	Total	474 (69.91%)	
	No total	204 (30.09%)	
Reoperation needed	Yes	176 (25.95%)	0.002
	No	502 (74.04%)	
Histological type	Papillary	579 (85.39%)	-
	Follicular	31 (4.57%)	
	Medullary	24 (3.53%)	
	Undifferentiated	14 (2.06%)	
	Lymphoma	12 (1.76%)	
	Secondary	10 (1.47%)	
	Squamous cell	4 (0.58%)	
	Sarcoma	3 (0.44%)	
	Myeloma	1 (0.14%)	
pTNM stage	I	501 (73.89%)	<0.0001
	II	90 (13.27%)	
	III	42 (6.19%)	
	IV	45 (6.63%)	
pT	pT1a	256 (37.75%)	
	pT1b	276 (40.70%)	
	pT2	78 (11.50%)	
	pT3	24 (3.54%)	
	pT4a	16 (2.36%)	
	pT4b	26 (3.83%)	
	pTm	2 (0.29%)	
pN	pN0	427 (62.97%)	
	pN1a	184 (27.14%)	
	pN1b	35 (5.16%)	
	pNx	32 (4.72%)	
pM	pM0	568 (83.78%)	
	pM1	46 (6.78%)	
	pMx	64 (9.43%)	

**Table 2 cancers-16-02019-t002:** Capsular and vascular invasion according to pN staging for well-differentiated TC (PTC and FTC). The *p*-values were calculated using Pearson’s correlation test, where *p* < 0.0001 indicates a statistically significant correlation between variables.

Feature	Number of Patients	*p* Value
pN0	384 (62.95%)	
Capsular invasion	142 (36.98%)	<0.0001
Capsular invasion absent	242 (63.02%)	
Vascular invasion	210 (54.69%)	
Vascular invasion absent	174 (45.31%)	
pN1a	166 (27.21%)	
Capsular invasion	129 (77.71%)	<0.0001
Capsular invasion absent	37 (22.29%)	
Vascular invasion	125 (75.30%)	
Vascular invasion absent	41 (24.70%)	
pN1b	31 (5.09%)	
Capsular invasion	31 (100%)	<0.0001
Capsular invasion absent	0 (0%)	
Vascular invasion	31 (100%)	
Vascular invasion absent	0 (0%)	
pNx	29 (4.75%)	
Capsular invasion	13 (44.83%)	<0.0001
Capsular invasion absent	16 (55.17%)	
Vascular invasion	13 (44.83%)	
Vascular invasion absent	16 (55.17%)	

**Table 3 cancers-16-02019-t003:** Relationship between capsular and vascular infiltration and hypoechogenicity in well-differentiated TC. The *p*-values were calculated using Pearson’s correlation test, where *p* < 0.0001 indicates a statistically significant correlation between variables.

Feature	Number of Patients	*p* Value
Hypoechogenicity observed	455 (74.59%)	
Capsular invasion	197 (43.30%)	<0.0001
Capsular invasion absent	258 (56.70%)	
Vascular invasion	198 (43.52%)	
Vascular invasion absent	257 (56.48%)	
Hypoechogenicity not observed	155 (25.41%)	
Capsular invasion	14 (9.03%)	<0.0001
Capsular invasion absent	141 (90.97%)	
Vascular invasion	14 (9.03%)	
Vascular invasion absent	141 (90.97%)	

## Data Availability

The datasets used and/or analysed during the current study are available from the corresponding author upon reasonable request.

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
