# Peer review of "Correlation of pN Stage and Hypoechogenicity with Tumour Encapsulation and Vascular Invasion in Thyroid Cancer (TC): A Comprehensive Analysis and Clinical Outcomes"

_cancers, 2024, doi:10.3390/cancers16112019_

Round 1

Reviewer 1 Report

Comments and Suggestions for Authors

The authors presented an interesting paper regarding the correlation between the pN stage and hypoechogenicity with tumor Encapsulation and Vascular Invasion in Thyroid Cancer. The manuscript is well written but the authors should better describe which are the main findings of this study: since the authors did not present any data on prognosis of these patients it is difficult to assess the real clinical impact of these findings and we may only speculate that patients with an higher rate of vascular invasion and tumor encapsulation will have a worse prognosis. Moreover, I would ask to authors to better detail how these patients were included: have all patients undergone total thyroidectomy and neck dissection?. Do you perform prophylactic neck dissection in all patients with TC? How did you select patients needing lymphadenectomy ?

Moreover, how do you evaluate the rate of hypoechogenicity ? Did you find a possible correlation between the pN status and sonographic patterns?.

Again, the authors should better describe how the interesting findings of this study could modify or improve the clinical and therapeutic management of patients with thyroid cancer. 

Author Response

We have referred to the recommendation in the attached file. 

Reviewer 2 Report

Comments and Suggestions for Authors

In this retrospective study of 678 thyroid cancer patients, the relationship between pN stage and ultrasound findings such as tumor encapsulation and vascular invasion was examined. The study found significant associations between advanced pN stage and the presence of encapsulation and vessel invasion, highlighting the potential of ultrasound features in predicting tumor behavior and emphasizing the importance of accurate N staging in thyroid cancer management.

-          The majority of pN1a patients exhibited encapsulation and vascular invasion. We cannot exclude that lymph-node involvement and activation of immune system may promote vascular invasion through the production of VEGF. Indeed, Gulcelik et al. reported higher VEGF expression in adjacent non-tumoral tissue in PTC patients with lymphocytic thyroiditis than in those without. In addition, Tian et al. reported that VEGF-C protein expression was significantly more frequent in papillary thyroid cancers with lymph-node metastases than without metastases. This topic and related mechanisms should be reported in the discussion section. Please, see and cite PMID: 35775885, 18652765.

-          It would be interesting to include in the analyzed data the presence or absence of chronic thyroiditis (biochemical or at least an ultrasound pattern).

-          The authors should better underline the main novelty of the present manuscript compared to previous studies.

-          The retrospective design of the study represents a major limitation. Limitations of this study should be reported in the discussion section.

-          The authors should better develop the future perspectives of this study.

-          Tables 1 and 2: please specify in the legend what p values refer to.

Author Response

(The authors gave the same response as above.)

Reviewer 3 Report

Comments and Suggestions for Authors

It is my pleasure to review the manuscript. The authors reported on the Correlation of pN Stage and Hypoechogenicity with Tumour Encapsulation 

and Vascular Invasion in Thyroid Cancer (TC): A Comprehensive Analysis and Clinical Outcomes. 

I think it is wonderful that there are so many cases. However, this paper in my opinion of poor quality. 

Major comments

  1. The analysis showed that pN stage is related to tumor encapsulation and vascular invasion, but I do not know how the results will change the actual clinical treatment strategy, prognosis, or benefit the patients.

  1. I do not know the definition of hypoechogenicity. Who determines it and how? In cases such as those with Hashimoto's disease, normal thyroid is also hypoechogenic, and I feel that the reproducibility is low.

  1. Since other pathologies may change the treatment strategy, it would be better to unify papillary carcinoma.

Major comment

Figure 2 should show the percentage on the vertical axis.

Author Response

(The authors gave the same response as above.)

Round 2

Reviewer 1 Report

Comments and Suggestions for Authors

The authors addressed all my  concerns

Reviewer 3 Report

Comments and Suggestions for Authors

The manuscript has been revised well. I think this manuscript is now acceptable.